# Taxon-specific aerosolization of bacteria and viruses in an experimental ocean-atmosphere mesocosm

Jennifer M. Michaud[1], Luke R. Thompson [2,3,4], Drishti Kaul[5], Josh L. Espinoza[5], R. Alexander Richter[5], Zhenjiang Zech Xu[2], Christopher Lee[1], Kevin M. Pham[1], Charlotte M. Beall[6], Francesca Malfatti[6,7], Farooq Azam[6], Rob Knight [2,8,9], Michael D. Burkart [1], Christopher L. Dupont[5] & Kimberly A. Prather[1,6]

Ocean-derived, airborne microbes play important roles in Earth's climate system and human health, yet little is known about factors controlling their transfer from the ocean to the atmosphere. Here, we study microbiomes of isolated sea spray aerosol (SSA) collected in a unique ocean–atmosphere facility and demonstrate taxon-specific aerosolization of bacteria and viruses. These trends are conserved within taxonomic orders and classes, and temporal variation in aerosolization is similarly shared by related taxa. We observe enhanced transfer into SSA of Actinobacteria, certain Gammaproteobacteria, and lipid-enveloped viruses; conversely, Flavobacteriia, some Alphaproteobacteria, and *Caudovirales* are generally under-represented in SSA. Viruses do not transfer to SSA as efficiently as bacteria. The enrichment of mycolic acid-coated Corynebacteriales and lipid-enveloped viruses (inferred from genomic comparisons) suggests that hydrophobic properties increase transport to the sea surface and SSA. Our results identify taxa relevant to atmospheric processes and a framework to further elucidate aerosolization mechanisms influencing microbial and viral transport pathways.

[1] Department of Chemistry and Biochemistry, University of California San Diego, La Jolla, CA 92093, USA. [2] Department of Pediatrics, University of California San Diego, La Jolla, CA 92093, USA. [3] Department of Biological Sciences and Northern Gulf Institute, University of Southern Mississippi, Hattiesburg, MS 39406, USA. [4] Ocean Chemistry and Ecosystems Division, Atlantic Oceanographic and Meteorological Laboratory, National Oceanic and Atmospheric Administration, stationed at Southwest Fisheries Science Center, La Jolla, CA 92037, USA. [5] J. Craig Venter Institute, La Jolla, CA 92037, USA. [6] Scripps Institution of Oceanography, La Jolla, CA 92037, USA. [7] Istituto Nazionale di Oceanografia e di Geofisica Sperimentale, Trieste, Italy. [8] Department of Computer Science and Engineering, University of California San Diego, La Jolla, CA 92093, USA. [9] Center for Microbiome Innovation, University of California San Diego, La Jolla, CA 92093, USA. These authors contributed equally Drishti Kaul, Josh L. Espinoza. Correspondence and requests for materials should be addressed to M.D.B. (email: mburkart@ucsd.edu) or to C.L.D. (email: cdupont@jcvi.org) or to K.A.P. (email: kprather@ucsd.edu)

A fundamental understanding of how bacteria and viruses become airborne remains elusive yet is central to understanding their role in atmospheric processes and the spread of disease. Microbial transmission and survival are influenced by particle size, relative humidity, temperature, environmental composition, and virus type[1–3], yet the mechanisms controlling their initial airborne transmission are poorly understood. The atmosphere contains vast bacterial ($6 \times 10^4$–$1.6 \times 10^7$ cells m$^{-3}$) and viral populations arising from a wide range of aquatic, terrestrial, and organismal biomes[4–7]. As the ocean covers 71% of the planet and contains 60–90% of the world's prokaryotes by cell abundance in open ocean and sediment[8], sea spray aerosol (SSA) represents a significant yet largely understudied source of airborne bacteria and viruses. Bacteria in aerosols can travel as far as 11,000 km, with air residence times of days to weeks[1,5,9], and algal viruses maintain infectivity over several hundred kilometers[10]. Airborne bacteria and viruses influence climate by serving as cloud seeds and inducing ice nucleation[11–14]. Microbes detected in clouds and precipitation have major impacts on ice formation and precipitation efficiency, yet their source remains largely unknown[15]. Airborne microbes also impact air quality through transmission of allergens and pathogens[16]. However, identification of species-level contributions of ocean microbes to atmospheric processes and air quality has never been definitely determined.

The production of SSA is a key process connecting the microbiome of seawater to the atmosphere. SSA is formed through bubble bursting occurring in the sea surface microlayer (SSML), the top 1–1000 μm of seawater. The SSML can act as a concentration zone for bioorganic molecules, microbes, and viruses[17,18] prior to their release as SSA;[19–23] however, oceanic observations of bacterial SSML enrichment are inconsistent[24–26] suggesting unknown influences. Jet drops containing bulk seawater can also transfer microbes from the ocean to the atmosphere[27]. Cultivation-based studies have observed differential transfer of bacteria to SSA in lab-scale systems[28,29]. While these studies did not access microbial diversity or produce natural SSA, they suggest taxon-specific aerosolization dynamics.

Studying aerosolization in natural systems is prohibitively difficult because confounding factors such as ocean and atmospheric circulation patterns prevent deconvolution of terrestrial and marine sources of airborne microbes[30]. Until now, most laboratory generation systems that produce isolated sea spray aerosols have not appropriately replicated natural aerosolization processes[31]. Accordingly, an ocean–atmosphere facility, developed collaboratively by chemists, oceanographers, and marine biologists was employed to study microbial aerosolization in a controlled setting using locally derived seawater and breaking waves[32]. This 13,000 L enclosed wave channel system is supplied with filtered air and is conducive to microbial growth at ocean-relevant abundances. It was verified experimentally to produce isolated SSA with the same size distributions, composition, and properties as those occurring in the atmosphere[32].

Here we examine if the transfer of bacteria and viruses from the ocean to the atmosphere is taxon specific and sensitive to environmental changes in the simulated natural ecosystem. For this purpose, we characterize bacterial and viral communities sampled from bulk (subsurface), SSML, and SSA at several time points during a 34-day nutrient-induced bloom event using read- and assembly based metagenomic approaches. We report SSA enhancement patterns correspond to related taxa and therefore, are determined at the genomic level, and that bacteria are generally more enriched in SSA than viruses. Changes to SSA enrichment as apparent responses to environmental cues or other signals are shared between related taxonomic classes and orders. This identifies taxa relevant to atmospheric processes and airborne systems. These relationships allow comparisons within taxa to determine aerosolization mechanisms with potential application to the understanding of environmental microbes and pathogen transmission.

## Results

**Mesocosm phytoplankton blooms**. Aerosol transfer of bacteria and viruses was monitored over 34 days of nutrient-stimulated phytoplankton blooms in a 13,000 L mesocosm. Filtered (50 μm) coastal Pacific seawater (32° 52′ 01.4″ N 117° 15′ 26.5″ W) was supplemented with nitrate, phosphate, and trace metals to induce phytoplankton growth under continuous illumination (45 μE m$^{-2}$ s$^{-1}$) with mixing provided by wave breaking or bubbling (Methods). Exogenous organic matter was not added. The conditions generated two phytoplankton blooms and corresponding heterotrophic bacterial blooms (Fig. 1a). Growth was monitored using chlorophyll *a* fluorescence to track bulk phytoplankton abundance and cell and virus counts in bulk seawater, SSML, and SSA (Fig. 1a, Table 1). Bulk chlorophyll *a* levels, cell counts by flow cytometry, and viral counts by epifluorescence microscopy were representative of natural oceanic bloom timescales and concentrations[8,33,34] (Supplementary Note 1). Due to the proximity of the SSA collection ports to the breaking waves, the average aerosol bacterial concentrations ($1.26 \times 10^7 \pm 0.19 \times 10^7$ cells m$^{-3}$) were 1000-fold higher than those estimated in natural environments where aerosols are sampled many meters above the sea surface and diluted by the atmosphere[9]. Aquatic bacterial abundance moderately increased with phytoplankton populations during the first phytoplankton bloom, while the peak for the second bacterial bloom was both larger in intensity and slightly delayed from the phytoplankton peak. The concentrations of bacterial cells in the bulk and SSML were positively correlated, while aerosol abundances were negatively correlated with seawater population. Bacteria were not immediately released into SSA from SSML, suggesting dynamics that depend on growth and accumulation in the bulk seawater and SSML prior to atmospheric release. In the ocean, virus-like particle concentrations are generally 5–25 times greater than bacteria[35], and here we observed ten-fold greater bulk viral concentration by cell (Fig. 1a) and particle counts (Table 1). However, average SSA concentrations of viruses ($1.19 \times 10^7 \pm 0.58 \times 10^7$ cells m$^{-3}$) and bacteria were nearly equivalent indicating preferential aerosolization of bacteria relative to viruses (Table 2).

**Genome assembly and taxonomic classification**. Bacterial and viral genomes were identified in bulk seawater, SSML, and SSA at six time points over the course of the experiment using size-fractionated metagenomics (Fig. 1, Supplementary Fig. 1). Following sequencing, both read-based (Kraken[36]) and assembly based approaches[37,38] were used to taxonomically characterize microbial communities in the bulk, SSML, and SSA (Supplementary Note 2, Supplementary Table 1-5). Comparisons between these methods were used to substantiate read-based annotations and aerosolization trends. For data reporting on species identity, additional spatial coverage analysis of genomes with full reference genomes was conducted (Supplementary Table 3). Genomes below a spatial coverage threshold (0.1% of total genome length) were removed. Assembly of k-mer and coverage-based binned contigs resulted in 24 draft genomes (Supplementary Table 4, 5). These assemblies were also cross-annotated using Kraken, with similar phylogenetic results based on whole genome and read analyses (Supplementary Table 6).

While unbiased assessments of microbial diversity in isolated SSA to date are lacking, phytoplankton bloom-associated copiotrophs—Alphaproteobacteria (Roseobacters), Flavobacteriia,

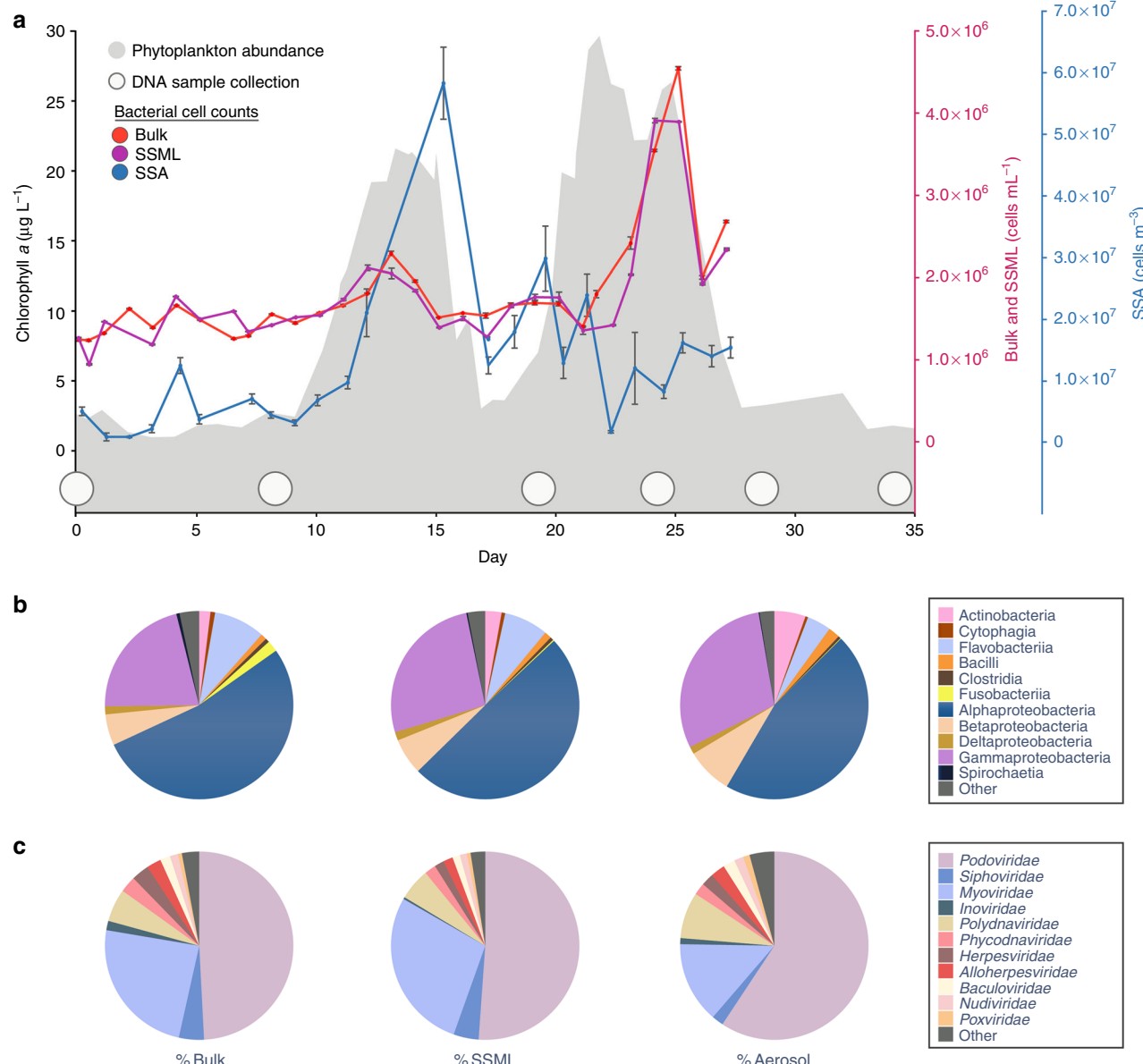

**Fig. 1** Bacterial and viral abundance across phytoplankton blooms. **a** Phytoplankton abundance is indicated by chlorophyll *a* fluorescence (gray). Bacterial cell counts by flow cytometry are shown for bulk (red), sea surface microlayer (SSML) (violet), and sea spray aerosol (SSA) (blue). $1 m^3 = 10^6$ mL of air volume. Bulk and SSML counts were performed in triplicate. Only single measurements were possible for aerosol samples. Error bars indicate s.d. and estimated error in aerosol measurements. Time points for taxonomic analysis are indicated by white circles. **b** Average percent of dominant bacteria classes and **c** average percent of dominant virus families during the blooms are shown for bulk, SSML, and SSA. Identities are from read-based analysis

| Table 1 Virus-like particle counts by microscopy in bulk, SSML, and aerosol samples | | | | | | | | | | | | | |
| --- | --- | --- | --- | --- | --- | --- | --- | --- | --- | --- | --- | --- | --- |
| Day | Bulk ($\times 10^7$) (vlp mL$^{-1}$) | Day | Bulk ($\times 10^7$) (vlp mL$^{-1}$) | Day | Bulk ($\times 10^7$) (vlp mL$^{-1}$) | Day | SSML ($\times 10^7$) (vlp mL$^{-1}$) | Day | SSML ($\times 10^7$) (vlp mL$^{-1}$) | Day | Aerosol ($\times 10^7$) (vlp mL$^{-1}$) | Day | Aerosol ($\times 10^7$) (vlp mL$^{-1}$) |
| 4 | 1.86 ± 0.40 | 13 | 1.63 ± 0.34 | 23 | 4.76 ± 0.65 | 7 | 1.32 ± 0.19 | 22 | 3.04 ± 0.43 | 0 | 1.58 ± 0.18 | 14 | 1.32 ± 0.15 |
| 5 | 1.00 ± 0.29 | 14 | 1.44 ± 0.26 | 24 | 3.63 ± 0.52 | 8 | 1.21 ± 0.22 | 23 | 3.66 ± 0.35 | 1 | 0.70 ± 0.08 | 18 | 1.39 ± 0.15 |
| 6 | 2.17 ± 0.52 | 15 | 1.72 ± 0.33 | 25 | 8.01 ± 0.80 | 9 | 1.21 ± 0.22 | 24 | 3.96 ± 0.53 | 2 | 0.77 ± 0.09 | 20 | 0.33 ± 0.04 |
| 7 | 1.68 ± 0.32 | 16 | 2.99 ± 0.36 | 26 | 5.50 ± 0.74 | 10 | 1.43 ± 0.28 | 29 | 5.56 ± 0.92 | 4 | 1.01 ± 0.11 | 22 | 0.55 ±± 0.06 |
| 8 | 0.60 ± 0.10 | 17 | 2.36 ± 0.26 | 27 | 5.07 ± 0.59 | 16 | 2.08 ± 0.27 | 32 | 1.59 ± 0.54 | 6 | 2.33 ± 0.26 | | |
| 9 | 1.93 ± 0.21 | 18 | 2.91 ± 0.42 | 28 | 5.26 ± 0.62 | 17 | 3.54 ± 0.42 | 33 | 2.35 ± 0.54 | 7 | 1.04 ± 0.12 | | |
| 10 | 1.10 ± 0.38 | 19 | 4.56 ± 0.43 | | | 18 | 3.85 ± 0.51 | 34 | 3.48 ± 0.68 | 9 | 2.14 ± 0.24 | | |
| 11 | 1.01 ± 0.23 | 21 | 4.36 ± 0.36 | | | 20 | 3.66 ± 0.57 | 35 | 4.06 ± 0.49 | 11 | 1.14 ± 0.13 | | |
| 12 | 1.24 ± 0.27 | 22 | 7.97 ± 0.79 | | | 21 | 3.90 ± 0.42 | 36 | 2.95 ± 0.34 | 13 | 1.20 ± 0.13 | | |

Error reported is s.d. and estimated error in aerosol measurements
*SSML* sea surface microlayer, *vlp* virus-like particle

and Gammaproteobacteria[39–41]—were abundant in bulk, SSML, and aerosol samples during the experiment. Comparisons between the taxa in these compartments reveal key differences in community distributions (Fig. 1b, c). Likely due to adaptation to the growth system and preference for species thriving in eutrophic conditions, the aquatic microbial population shifted significantly from the original oceanic composition, but after Day 8 remained relatively constant for the duration of the experiment with small shifts in individual taxa similar to bacterial community shifts observed during natural marine phytoplankton blooms[39,42] (Supplementary Fig. 1a). The blooms also contained taxa not often found in seawater including an avian strain of *Escherichia coli* and a novel strain of *Legionella*, as revealed by assembly of nearly complete genomes (Supplementary Table 4, 5). The novel *Legionella* strain has 70–80% amino acid identity to orthologs in *Legionella pneumophila* (Legionnaires' disease) and *Legionella drancourtii* and has both Type II and Type IV secretion systems typical of pathogenic strains. *E. coli* and *Legionella* were present in initial samples (Supplementary Fig. 2), suggesting they were native to the coastal community but became enriched in our system during the experiment. The recovery of these genomes is consistent with increasing evidence for enteric contaminants in coastal marine waters[43,44].

*Podoviridae* and *Myoviridae* were the most common viruses identified in raw read assignments, and the viral population varied across the air–sea interface (Fig. 1c). Despite recent advances in characterization of marine viral genomes[45], fewer viral genomes were recovered compared to bacterial genomes due to under-developed viral databases. Populations shifted slightly from the initial community and were more dynamic during the bloom than the bacterial populations (Supplementary Fig 1b). Draft genomes were recovered for five bacteriophage as well as an nucleoplasmic large DNA virus (NCLDV) (Supplementary Table 4, 5). While analyses were not targeted toward eukaryotic genomes, the recovery of a draft genome for a diatom chloroplast and parallel microscopic observations of a small centric diatom suggest diatoms are the source of the phytoplankton blooms.

**Bacterial aerosolization**. To examine the transfer of annotated genomes across the air–sea interface, we compared relative abundances in bulk seawater, SSML, and SSA. The relative enrichment occurring in the sea-to-air exchange process was determined by calculating the aerosolization factor (AF) for each genome in the 0.2–3 µm size fraction for bacteria and 0.025–0.2 µm for viruses. This is defined as the ratio of the fraction in SSA (A) to the fraction in bulk seawater (B) or the fraction in the SSML (S) (Eq. 1). Surface enrichment factors are calculated similarly using the ratio of S to B (S:B).

$$\text{AF(Aerosolization factor)} = \frac{\text{Species fraction in aerosol (A)}}{\text{Species fraction in bulk (B) or SSML (S)}}$$
(1)

Genomes that contributed <0.01% to the population in any sample or <0.1% in more than half of samples were omitted to avoid erroneous calculations from genomes below threshold abundances. The AFs for bacteria and viruses over the course of the bloom are shown in Fig. 2. The top-right quadrants (red) of Fig. 2 indicate genomes enriched in aerosol relative to both bulk and SSML (A:B and A:S > 1); the bottom-left quadrants (blue)

| Table 2 The ratio of aerosol to bulk virus and bacteria counts by flow cytometry and microscopy | | |
|---|---|---|
| **Day** | **Virus aerosol: bulk** | **Bacteria aerosol: bulk** |
| 4 | 0.54 | 11.39 |
| 7 | 0.62 | 9.75 |
| 9 | 1.11 | 3.65 |
| 11 | 1.13 | 8.87 |
| 18 | 0.48 | 16.30 |
| 20 | 0.07 | 11.56 |
| 22 | 0.07 | 1.42 |
| Average | 0.68 | 11.13 |

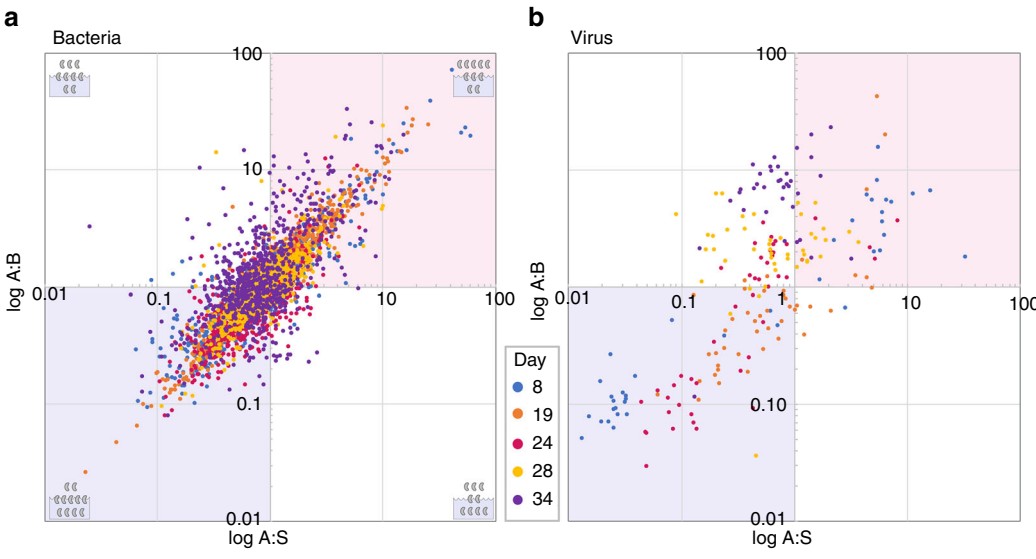

**Fig. 2** Aerosolization of bacterial and viral genomes across the blooms. The ratio of fraction of genomes in aerosols to the fraction in bulk (A:B) plotted against ratio of fraction of genomes in aerosols to those in sea surface microlayer (A:S) is shown for bacteria (**a**), and viruses (**b**), over the course of the experiment. The upper right quadrants (red) of **a** and **b** indicate genomes enriched in aerosol relative to both bulk and SSML (A:B and A:S > 1); the lower left quadrants (blue) indicates genomes that are primarily waterborne (A:B and A:S < 1). The upper left quadrants indicates genomes enriched relative to bulk but not SSML (A:B > 1, A:S < 1), and the lower right indicates species enriched in aerosol relative to the surface but not bulk (A:B < 1, A:S > 1). The data sets represent **a** 700 bacterial genomes and **b** 46 viral genomes identified by read-based taxonomic assignments filtered of genomes below abundance thresholds. Graphics in the corners various quadrants in **a** show examples of expected species ratios in indicated portion of graph

indicate genomes that are primarily waterborne (A:B and A:S < 1). The upper-left quadrants indicate species enriched relative to bulk but not SSML (A:B > 1, A:S < 1), and the bottom-right indicates species enriched in aerosol relative to the SSML but not bulk (A:B < 1, A:S > 1). A Gaussian distribution of species, centered at (1,1), would be expected if aerosolization was random. However, the "spindle"-shaped distribution of the data reveals non-random patterns. The distribution is similar in the genome spatial coverage trimmed data set (Supplementary Fig 3a, b). Histograms and normality analysis revealed positively skewed data with a non-normal distribution and mean bacterial AF of 0.90 (Supplementary Fig. 4a-c, Supplementary Table 7). Genomes predominantly exhibit SSA enrichment or depression with respect to both bulk and SSML. While water-enriched species (AF values <1) represented 52% of the population, aerosol enrichment was much larger in amplitude than diminishment, reflected in a greater extension of data points into the red region compared to the blue region of Fig. 2a. The average maximum AF value is two times larger in amplitude than the average minimum value.

A genetic signal in bacterial AF is evident at fine-scale taxonomic levels when individual genomes are compared (Fig. 3a, Supplementary Figs. 5–8). Closely related species have nearly identical aerosolization patterns that are propagated across environmental and enrichment changes. Genomes within the same class (Actinobacteria and Flavobacteriia) or order (Rhodobacterales, Sphingomonadales, and Alteromonadales) typically have similar aerosolization profiles. Examples of bacterial genomes with consistently high AF values include *Rhodococcus erythropolis*, *Cutibacterium acnes*, and *Methylobacterium radiotolerans*. Conversely, genomes such as *Flavobacterium indicum*, *Cellulophaga lytica*, and *Erythobacter litoralis* had persistently low AFs. The low AF values of the *Erythobacter* draft genome provide ancillary assembly based evidence for the read-based analysis (Supplementary Fig. 7).

Comparisons of temporal phylogenetic patterns provide critical information needed to determine aerosolization mechanisms, especially in rare cases where a species' aerosolization departs from other members of its genus, order, or class (Fig. 3a, i–ii). For example, *Corynebacterium kroppenstedtii* and *Corynebacterium aurimucosum* are the same genus yet have opposite aerosolization patterns (Fig. 3a, i). Likewise, the low AF values of *Moraxella catarrhalis* contrast with other more aerosolized Pseudomonadales (Fig. 3a, ii). Direct observations and in silico compositional, genomic, and metabolomic studies guided by divergent aerosolization patterns of closely related genera will provide a biochemical basis for aerosolization biases further discussed below.

**Viral aerosolization**. Within our experimental scheme, viruses did not aerosolize as efficiently as bacteria. The ratio of average bacterial counts in aerosol (cells m$^{-3}$) to bulk (cells mL$^{-1}$) was 11.13, while the average free virus counts in aerosol (particles m$^{-3}$) to bulk (particles mL$^{-1}$) was 0.68 (Table 2). In contrast, SSML enrichment is approximately the same for both (0.98 and 1.02 for bacteria and viruses, respectively). In read-based sequence analysis, the proportion of viral to bacterial reads in aerosol at the division level was consistently lower than bulk in all size fractions suggesting decreased viral transfer even when they are associated with larger particles such as bacteria or phytoplankton (Supplementary Table 8). Decreased viral transfer is further supported by comparisons of data distributions of viral and bacterial communities in read-based analyses (Supplementary Fig. 4a-f, Supplementary Table 7). Viral distributions were shifted more negatively and had lower mean AFs than bacterial populations (viral—0.72 ± 4.70, bacterial—0.90 ± 2.32). Studies in pathogen transmission suggest that smaller bioaerosol size leads

to increased residence time in the air[1,2]. However, these results indicate that size is not the only contributor to marine viral aerosolization.

Despite depressed aerosolization relative to bacteria, differential transfer of select and functionally specific viral genomes into SSA was observed (Fig. 2b, Supplementary Fig. 3b). However, the viral AF factors cluster around different values according to sampling date, indicating broad environmental parameters affect the aerosolization of most viruses; bacterial AFs demonstrate a more linear relationship less dependent on environmental variance (Fig. 2, Supplementary Fig. 4b, e). The temporal clustering is further observed in shared aerosolization by most taxa on a given day as seen by bands of similar AF intensity in Fig. 3b.

Taxon-specific patterns were also detected; however, aerosolization differences seem to delineate across the broader physiological trait of viral envelopes in addition to taxonomic order (Fig. 3b, Supplementary Figs. 6–7, 9). Non-enveloped viruses are less enriched in aerosol compared to lipid-enveloped species. The tailed bacteriophage, *Caudovirales* (*Myoviridae*, *Podoviridae*, and *Siphoviridae*), were generally waterborne as were phage identified in the binned assemblies (Supplementary Fig. 7); lipid-enveloped *Polydnaviridae* and *Alloherpesviridae* were enriched in aerosol. Similar to bacteria, while most viral genomes followed the aerosolization patterns of their taxa, select genomes differed from their closest relatives in their aerosolization behavior revealing genomic differences that may be useful in determination of relevant biochemical pathways. *Ostrecoccus lucimarinus* virus 1 (Fig. 3b, i) was greatly diminished in aerosol unlike its relatives, while *Pseudomonas* phage LKA1, *Klebsiella* phage KP15, and *Klebsiella* phage KP27 (Fig. 3b, ii–iii) were consistently aerosol enriched in comparison to the other bacteriophages. These results indicate that many viruses have aerosolization patterns governed primarily by environmental conditions and influenced by general morphological characteristics.

**Temporal dynamics**. As observed in the AF values of individual genomes, some bacteria and viruses demonstrated consistent behavior (enhanced or diminished) during the blooms, while others exhibited intermittent aerosolization enrichment patterns (Fig. 4, Supplementary Figs. 5–9). Constitutive aerosolization, defined here as when 80% of a species' observed AF values were either all greater or less than 1, was present in 45% of the bacterial genomes, and 35% of viruses. Constitutive aerosolizers like *R. erythropolis*, *F. indicum*, and *Pseudomonas* phage LKA1 likely have a surface composition that supports or inhibits aerosolization. Intermittent aerosolizers, such as *Dinoroseobacter shibae*, *Legionella* sp., or *Phaeocystis globosa* virus may undergo changes that alter physiological properties. For example, *L. pneumophila* is known to have both non-motile, non-pathogenic, and motile-pathogenic phases of its life cycle[46]. Thus, it could be postulated that such morphological changes dictate aerosolization, and intermittent release might promote host exposure.

Both constitutive and intermittent aerosolizers exhibit some variation in their AFs, indicating that environmental conditions influence aerosolization. The resulting changes in response to these external factors were also correlated across select taxa (Fig. 3). This coordination is seen in intermittent Alphaproteobacteria and high AF lipid-enveloped viruses that were generally more waterborne on Day 8 (pre-bloom) while several Gammaproteobacteria genomes were aerosol enriched. During the second bloom (Day 28), when total seawater bacterial counts were high, Flavobacteriia and nearly all viral genomes became enriched in SSA. Because viruses were generally more intermittent than bacteria and share aerosolization patterns across taxa during a

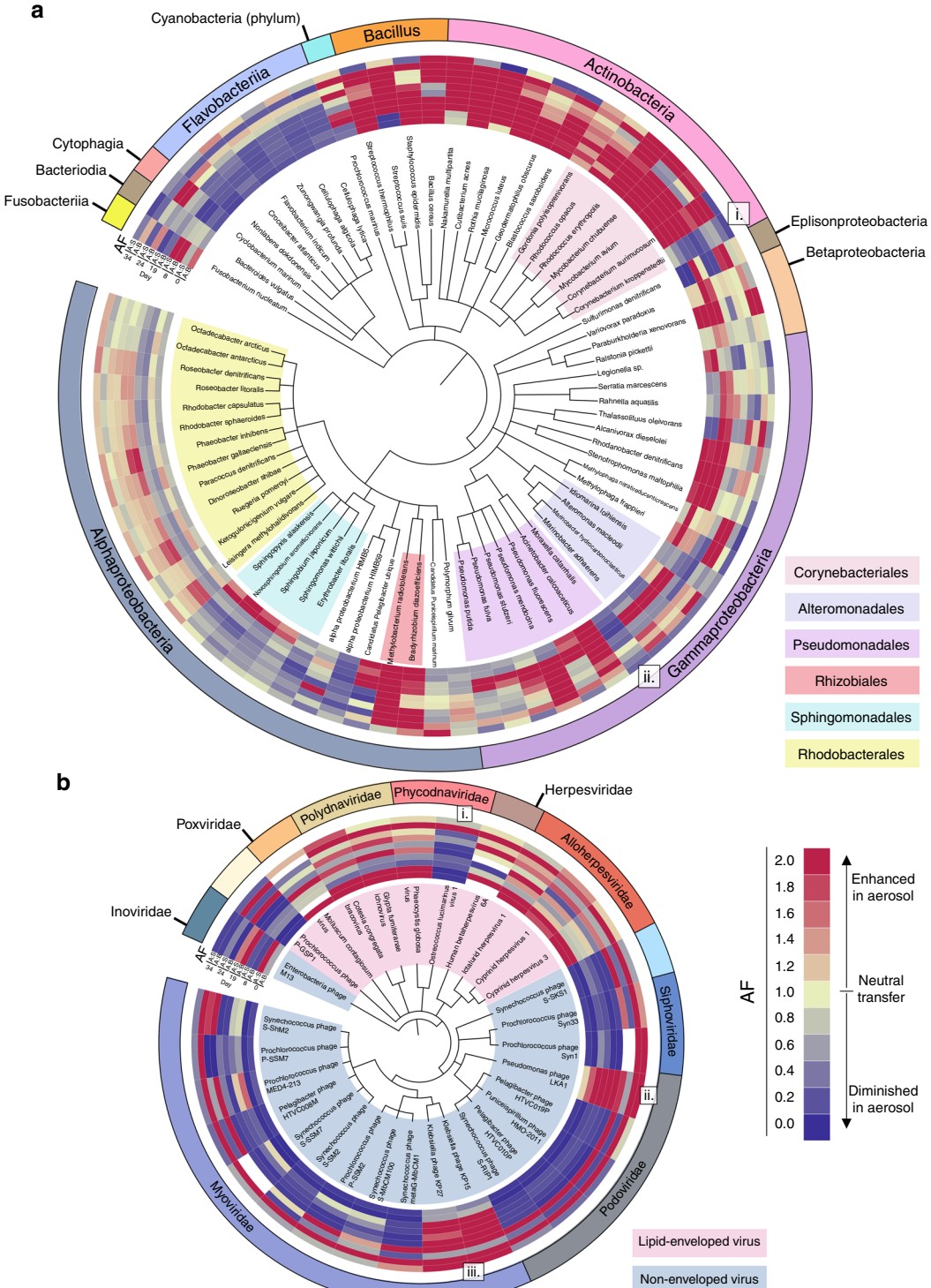

**Fig. 3** The taxonomic basis of bacterial and viral aerosolization. Genome-based phylogenetic trees from read-based annotations trimmed of genomes with low spatial coverage: **a** 76 bacterial genomes and **b** 30 viral genomes. Aerosolization factors (AF), A:B and A:S, on different days are indicated (Blue = Diminished aerosolization; Yellow = Neutral aerosolization; Red = Enhanced aerosolization). Blanks indicate samples below threshold limits. **a** Bacteria class is indicated on outer ring to further indicate species and shading of species names indicate orders of interest. In **b**, the outer ring denotes viral family and shading indicates presence or lack of a viral envelope. i, ii, and iii indicate genomes with aerosolization patterns that differ from their closest relatives. Trees generated using phyloT (http://phylot.biobyte.de/) and iTOL (http://itol.embl.de/)

given time period, viral aerosolization seems especially sensitive to chemical and biological water composition changes.

**Surface enrichment**. Taxon-specific variation in surface enrichment demonstrated factors that support SSML

accumulation also support aerosolization. Surface enrichment factors (S:B) predominately indicated SSML enrichment or neutral accumulation from bulk water (Supplementary Figs. 5, 7–9), and the magnitude of the S:B and AF values were positively correlated.

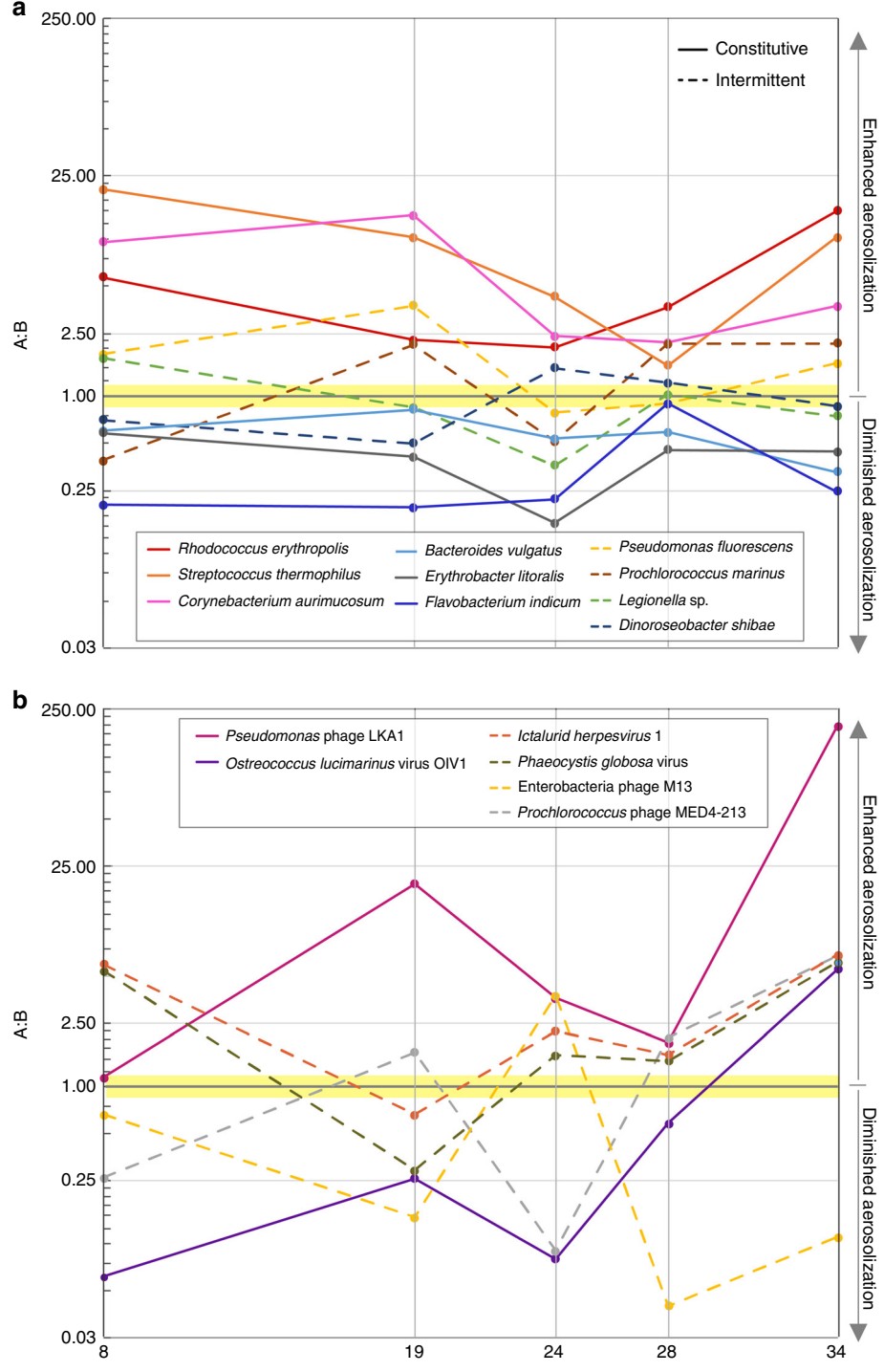

**Fig. 4** Temporal aerosolization dynamics. Intermittent versus constitutive aerosolization patterns are shown for select bacteria (10 genomes) **a**, and viruses (6 genomes) **b**, across the phytoplankton blooms. Solid lines indicate genomes that are always enriched or always diminished (constitutive). Dashed lines indicate intermittent genomes that demonstrate both states

The constitutive, low AF Flavobacteriia were the least surface enriched with an average S:B of 0.95 compared to 1.92 for all species examined (Supplementary Fig. 5). Constitutively high AF Actinobacteria had a corresponding above average S:B of 3.92. Viruses were more surface enriched than bacteria with an average S:B of 4.70 with a large contribution from global viral surface enrichment on Days 28 and 34 (Supplementary Fig. 9). The surface enrichment of most viral taxa on these days corresponded to global viral A:B enrichment (Fig. 3b). Excluding these days, the average S:B falls to 1.70, low-AF-non-enveloped viruses have an average S:B of 1.70, and high-AF-enveloped viruses have an average S:B of 3.69. As SSML is organic enriched[17], hydrophobic cell surface properties likely promote both surface enrichment and aerosolization. This taxon-specific surface accumulation also helps account for the disparate oceanic observations of microbial sea surface enrichment[24–26].

## Discussion

Guided by these results, comparative genomics indicate cell surface modifications influence aerosolization. The Gram-positive Actinobacteria had high and constitutive AF values. Within this class, Corynebacteriales possess an unusual waxy coat consisting of mycolic acids made of ultra-long chain fatty acids ($C_{25-30,60-90}$)[47,48]. In contrast to the rest of its order, *C. kroppenstedtii* (Fig. 3a, i) lacks mycolic acids[49], and this missing hydrophobic cell envelope component may explain its reduced aerosolization relative to closely related strains.

The ability to rapidly desiccate and tolerate desiccation are important factors in viral pathogen transmission as they increase the time particles remain aloft and viable[50,51]. Most respiratory viruses, like influenza and corona viruses, are lipid enveloped and able to survive longer at low relative humidity (15–40%); non-enveloped viruses require a minimum of 70% humidity for viability[1,3]. Similar trends are echoed in these data (Fig. 3b). Tailed bacteriophage utilize surface capsid proteins that are likely hydrophilic, while herpesviruses are lipid enveloped[52–54]. Here, *Caudovirales* were generally waterborne while *Herpesvirales* were aerosol enriched. Hydrophobic surface features probably assist in rapid cell desiccation, which is one potential mechanism of aerosolization and surface enrichment.

However, hydrophobicity is likely not the only mechanism of aerosolization and SSML enrichment. *Bacillus* strains, generally encapsulated in hydrated polysaccharide coatings[55], also had consistently high AF and S:B values (Fig. 3a, Supplementary Fig. 5). Some Gammaproteobacteria in the order Pseudomonadales are observed with capsules. *M. catarrhalis* unlike its closest relatives had constitutive, low AFs (Fig. 3a, ii), and contradicting reports describe both the presence and lack of a cell capsule[56]. If different strains or growth conditions alter encapsulation states it could alter temporal aerosolization dynamics.

This study provides a needed baseline for experimental marine microbial aerosolization studies that may now incorporate new variables such as diurnal light cycling, zooplankton grazing, different seed populations, and alternative nutrient amendments to further evaluate aerosolization dynamics of specific species and environments. The results reveal the taxon-specific dynamics of viral and bacterial transfer across the air–sea interface during phytoplankton blooms, conditions that generate increased bacterial and viral water concentrations with increased potential impact to airborne environments. Aerosol enrichment is dictated by inherent biochemical properties and influenced by environmental conditions. Taxon-specific patterns are applied to reveal different aerosolization mechanisms and indicate trends of broad taxonomies applicable to a variety of systems. Taxa with enhanced aerosolization likely have a larger influence on climate properties, and studies with relevant isolates will clarify specific contributions to atmospheric processes. Several bacterial lineages of known respiratory and airborne pathogens are identified in this study, including *Legionella*, *E. coli*, *Corynebacterium*, and *Mycobacterium*. Factors that modulate SSA formation likely will apply to other aerosolization processes. Elucidating the specific aerosolization mechanisms of pathogens, either from environmental reservoirs or from hosts themselves, may form the basis of therapeutics that target aerosolization by interaction with surface features or inhibition of relevant biosynthetic pathways. These findings provide a framework to resolve such mechanisms.

## Methods

**Waveflume mesocosm bloom.** The wave channel was set up as described in Wang et al.[57]. 13,000 L of ocean water was collected off Scripps Pier in La Jolla, CA (32° 52′ 01.4″ N 117° 15′ 26.5″ W). Water was filtered using 50 μM Nitex mesh to remove grazing zooplankton before transfer into the wave channel tank. The wavechannel is a 33 m × 0.5 m × 1 m (0.6 m water depth), enclosed (supplied

HEPA-filtered air), glass, growth system equipped with a hydraulic pump attached to a vertical paddle at one end. Each pulse created one simulated wave that broke midway through the channel just upstream of sampling ports and is calibrated to mimic aerosol formation that occurs during natural wave breaking[32]. Aerosol sampling ports were positioned ~3 ft downfield of the general position of the breaking waves. Bloom conditions were initiated by the addition of F/2 medium (Proline, Aquatic Eco-Systems, Apopka, FL) supplemented separately with sodium metasilicate. F/2 is a common marine medium for algae cultivation that provides 880 μM $NaNO_3$, 36 μM $NaH_2PO_4 \cdot H_2O$, 70 μM $Na_2SiO_3 \cdot 9H_2O$ along with trace metals[58]. An additional 9 μM sodium phosphate was added to the wave flume on July 25, 2014 as phosphate was depleted in the system. No exogenous carbon or additional nutritional supplementation was given for bacterial cultivation. The tank was continuously illuminated (45 μE m$^{-2}$ s$^{-1}$). During periods when aerosols were not being collected, additional aeration was provided by bubbling with HEPA-filtered air through perforated tygon tubing at the bottom of the tank.

**Sample collection.** Samples were taken for DNA analysis on 6 days during the phytoplankton bloom from bulk water (2 L), SSML (200 mL), and aerosol (collected for 3 h at a flowrate of 450 L min$^{-1}$) resulting in 18 samples total. Bulk samples were collected via siphon from 3 ft below the water surface, and SSML samples were collected by the glass-plate method[59]. Aerosols were collected using a SpinCon® PAS 450-10A Portable Air Sampler (Scepter Industries) as described in Yooseph et al.[60]. Prior to each sample collection, the SpinCon® was disinfected by autoclaving removable parts, soaking and/or spraying parts non-detachable or non-removable components with 70% ethanol where appropriate, and using new anti-microbial tubing. The inlet was plumbed directly to the outlet of the wave flume. Aerosols for DNA analysis were concentrated into 7.5–10 mL of sterile PBS (pH 7.4) over 3 h of wave breaking. Samples for bacterial and viral counts were collected similarly, with the exceptions of aerosol, which were collected by impingement at 1 L min$^{-1}$ into sterile seawater. Samples for counts were immediately fixed in 0.05% glutaraldehyde, flash-frozen, and stored at −80 °C. A system flow variance of 10% was accounted for in aerosol standard deviations.

**Phytoplankton, cell and virus abundance.** Phytoplankton growth was monitored using a Turner AquaFluor fluorometer that measures in vivo fluorescence of chlorophyll *a*. Bacterial abundance was determined by flow cytometry at the University of Hawaii Department of Oceanography using Hoechst 3442 staining[61]. A Beckman–Coulter Altra flow cytometer (operated by the SOEST Flow Cytometry Facility, www.soest.hawaii.edu/sfcf) was connected to a Harvard Apparatus syringe pump for quantitative analyses. Free viral abundance was determined on filtered samples (0.02-μm Anodisc, Whatman Nuclepore®) using epifluorescence microscopy (Keyence BZ-X700) with SYBR Green-I (Thermo-Fisher) staining[62]. It was not possible to perform aerosol virus counts on replicate samples due to the small sample sizes determined by the parameters of the experiment and volumes required by other analyses. Standard deviation reported reflects variance in air flows.

**DNA extraction.** Immediately after collection, samples were size fractionated by serial filtration on 3-μm polycarbonate filters (Whatman Nucleopore®), then 0.2-μm polyethylsulphone filters (Pall Supor® 200), and then 0.025-μm mixed cellulose filters (MF-Millipore®). Full sample volumes (2 L bulk, 200 mL SSML, and 10 mL aerosol) were run through each filter with the exception of the 0.025-μm bulk samples for which only 1 L was filtered. Filters were stored at −80 °C until extraction. DNA was extracted from the membranes according to Boström et al.[63]. This included overnight lysis, phenol chloroform extraction, and coprecipitation with GlycoBlue™ (ThermoFisher). DNA concentration was quantified by Qubit™ (ThermoFisher). DNA was not amplified prior to sequencing.

**Metagenomic sequencing.** Libraries were prepared using a Nextera XT DNA Library kit (Illumina). Sequencing was performed at the Institute for Genomic Medicine at UC San Diego on a HiSeq 4000 DNA sequencer (Illumina). Paired reads were trimmed and quality-filtered using Trimmomatic 0.32[64] in paired-end mode with parameters ILLUMINACLIP:NexteraPE-PE.fa:2:30:10 LEADING:3 TRAILING:3 SLIDINGWINDOW:4:15 MINLEN:36.

**Taxonomic profiling of metagenomes.** Trimmed and quality-filtered metagenomic sequence reads were analyzed using Kraken[36] (v0.10.5) to determine the identity and abundance of eukaryotic, bacterial, archaeal, and viral species present. Kraken was run in paired mode with the full NCBI RefSeq database (release 70, April 30, 2015) using default cutoffs. Genus-level annotations in Kraken are nearly 100% reliable in benchmark studies when the genus is in the database, while species assignments are 89% accurate with human microbiome data sets[36]. This accuracy is much lower in environmental samples because the representation of relevant genomes in the database is much less complete. To prevent errors in calculations, the taxonomic data set generated with Kraken was filtered to exclude any species that had any sample containing <0.001% of the total population or if 10 or more samples (of 18 total) contained <0.01%. Spatial coverage analysis was performed to determine the percent of the genome that is represented in a reference genome. Spatial coverage for each genome was calculated by implementing the following

pipeline: (1) parsing Kraken label output files to generate individual fastq files for each taxonomy hit; (2) mapping the reads to their respective genome assembly (e.g., full chromosomes, contigs, plasmids, etc.) [HISAT2 v2.0.0][65]; (3) converting the *.sam files to *.bedgraph files [bedtools v2.24.0][66]; and (4) concatenating all sequences in the genome assembly and calculating the proportion of positions that were mapped. Eighty-five percent nucleotide identity is the mapping criteria. The availability of reference genomes allowed analysis of 214 of 243 genomes included in the trimmed set of 197 bacterial and 46 viral genomes. This represented 28% of the total genomes in the original abundance trimmed set (756 genomes). The coverage trimmed data set omitted species below 0.1% spatial coverage to avoid erroneous annotations due common elements such as transposons (Supplementary Table 3).

**Draft genome assembly**. Draft genomes were assembled using the pipeline described by Dupont et al.[37]. Sixty-two individual shotgun sequenced metagenomes were assembled separately using SPAdes (v3.8.0, --meta mode, default kmer sizes of k21, 33, 55)[67], with subsequent co-assembly of the assemblies using CLC (QIAGEN) to collapse redundant contigs. Genome quality throughout was assessed using QUAST(v3.0) with default settings[68]. Original reads for each sample were mapped to the contigs (CLC) to build sample specific coverage information. All contigs greater than 5 kbp were clustered into genome bins using VizBin[69] and completeness of each of these bins was assessed using CheckM (v1.0.5)[70] (Supplementary Table 4). Subsequently, 14 bins were recovered from a pangenome in the data by utilizing hierarchical clustering and sample specific coverage information. Genome annotations were determined utilizing APIS (https://github.com/jhbadger/APIS)[71] and manual curation using PhyloDB (https://scripps.ucsd.edu/labs/aallen/data)[71] as the reference database. APIS automates the process of sequence similarity, alignment, and phylogenetic inference for each protein in a given data set[72]. APIS results were manually curated to determine predicted bin level taxonomy. All recovered genomes were reported regardless of completeness. To assess agreement between assembly based and read-based annotations, assemblies were additionally annotated using the methods used in read-based analyses (Kraken with the NCBI Refseq database) (Supplementary Table 6).

**Data analysis**. AF was calculated as the ratio of the fraction of species in the aerosol compartment to either the bulk water or SSML compartment using both read-based phylogenies from Kraken and draft genome data. Surface enrichment factors were similarly calculated using species fractions ratios of SSML to bulk seawater. To make comparisons of individual species more manageable, the free bacteria data set was further trimmed. Species were sorted by AF and 20 species were selected at random for the following values: AF > 5, AF 1.5–5, AF 1 to 1.5, AF ≈ 1, AF 0.667–1, AF 0.20–0.667, AF < 0.2. To ensure all abundant species were included, this was amended with the top 50 abundant species resulting in 197 bacteria species. This set was further utilized for coverage analysis. Phylogenetic trees were constructed in PhyloT (http://phylot.biobyte.de/) and visualized in iTOL (http://itol.embl.de/)[73]. Species were annotated with corresponding AF data to compare phylogeny and aerosolization patterns.

**Data availability**. Sequence read and assembly data are available in the BioProject database via accession number PRJEB20421.

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

## Acknowledgements

This work was supported by the Center for Aerosol Impacts on Chemistry of the Environment (CAICE), an NSF Center for Chemical Innovation (CHE-1305427), NSF IOS 1516156 (to M.D.B.), and NSF OCE-1259994 (to C.L.D). C.L.D., D.K., J.L.E., and R.A.R. were supported by a grant from the Beyster Family Fund of the San Diego Foundation. The authors would like to thank Andrew Allen, Ariel Rabines, Kristen Jepsen, Karen Selph, Melissa Carter, Tom Hill, and CAICE members for their valuable contributions.

## Author contributions

J.M.M., K.A.P., C.L.D., M.D.B., F.M., and F.A. designed the research. K.A.P. is the director of CAICE and oversaw the large-scale wave channel experiment. J.M.M. and C.L. performed experiment. J.M.M. and K.M.P. prepared samples. C.L.D., L.R.T., D.K., J.L.E., R.A.R., Z.Z.X., and R.K. designed and conducted metagenomics analyses. J.M.M. conducted aerosolization data analyses. C.M.B. and F.M. performed viral counts. J.M.M., C.L.D., K.A.P., L.R.T., M.D.B., and R.K. prepared manuscript. Any opinions, findings, conclusions, or recommendations expressed in this article are those of the authors and do not necessarily reflect the views of the National Science Foundation.

## Additional information

**Competing interests:** The authors declare no competing interests.

