## [Peer Review File · Nature Communications]

Reviewers' comments:

Reviewer #1 (Remarks to the Author):

Title: Selective transfer of bacteria and viruses from the ocean to the atmosphere

Authors: Michaud et al.

Summary

This article presents a study on the transfer of bacteria and viruses from the ocean to the air during phytoplankton blooms. The authors find, interestingly, that bacteria and viruses are transferred to the air at ratios that do not always equal their concentrations in the upper layers of the fluid. I have a few minor comments below, but generally believe this is a valuable contribution. My biggest issue is that the manuscript doesn't spend much time discussing the details of the experimental setup, which is a simulated ocean rather than the real thing.

Specific Comments

1. Lines 36 – 38. The data suggests an influence of surface properties, but how do the authors quantify the hydrophobic properties of the bacteria? Are they simply inferred from what is known of the cell envelope?
2. Lines 103 – 105. Why is there such a big difference in concentration? Wouldn't the amount of bacteria in the sample be independent of the distance to the collection ports? Do the authors expect that the difference is due to changes in cell viability? On a related point, how is the number of cells quantified?
3. Lines 151 – 153. It would seem that the shift in microbial population is a problem, how do we know that some of the species that disappeared wouldn't be preferentially released? Do the authors have hypotheses for why the population changes so quickly?
4. Lines 159 – 161. On a related point as above, how much does the microbiome in the ocean change typically in the amount of time relevant for these experiments?
5. How does the simulated ocean work, hydrodynamically (e.g. wave generation and any mixing)? Would be helpful to see how the system mimics ocean hydrodynamic properties.
6. How do the taxa found in the aerosols compare with those from other studies?
7. Line 369 and Line 1: It seems that the title is too general, I think 'phytoplankton bloom' should be found in the title since the authors really explored a particular situation.
8. Lines 545 – 546. Wouldn't you expect more realistic microbiome performance if you integrate light and dark cycles? Further, I expect that the aeration also influences the aerosol process and the stability/composition of the microbiome.

Reviewer #2 (Remarks to the Author):

The authors have identified a critical and under-studied area of science, and this manuscript is generally well written.

However, I'm particularly concerned that readers would not be able repeat these analyses or sampling based upon the information provided, and the viral metagenomics sequencing needs to be clearly described as quantitative or not depending upon the type of amplifications used in the library preps. My concerns and suggestions are as follows.

1) Why was it impossible to count replicate samples for the viruses? Replication is critical for insuring good counts as these are notoriously problematic and variable.

2) The methods need more details. For DNA extraction it is not clear whether the authors captured viral fraction samples from 2L of seawater or 200ml of SSML onto 0.02um filters. This would be nearly impossible, so if not what were the volumes captured on those filters? As well, how much DNA resulted? This is critical as amplification is commonly needed for viral metagenomes, but many amplification methods render the data non-quantitative, which would be problematic for most of the virus data presented in this paper. The informatics write-ups need to include versions and parameter settings for programs and dates for database access so that analyses could be repeated. Without this information it is not clear how to interpret the findings.

3) All figures need to be legible – the fonts were far too small.

4) In Fig 1a there are no viral count data, in spite of suggestion at L115 that there is. This is worth showing in the main text since a major conclusion is drawn from these data (that viruses are less aerosolized).

5) In Fig 1c – what fraction of the reads were mapped to a reference genome? (reader is led to believe 100% with the plots shown) this is important for interpreting all of the downstream data presented on viruses.

6) L128 and L168 – not clear how complete a draft genome needs to be to be reported here, and how the cut-offs were chosen

7) What is spatial scaffold coverage?

8) L160 – where are the data that E coli and Legionella were in the original seawater? This is a really important control to understand whether there are not contamination issues in the system.

9) L168 – how many draft genomes were recovered?

10) The main text needs to include some information on the quality of the metagenomes, as well as specific comparison of read-based vs assembly-based analyses. As the authors state in the results, Kraken does well where the reference genome database is strong, but its "accuracy is much lower in environmental samples". This will be much worse for viruses since the databases are so bad. What are the cut-offs used to "map" a read to a reference genome? I would think utilizing the draft genomes assembled in main text figures would be needed to show this is a meaningful analysis since Kraken is not commonly used in environmental work (I see this is in L130 in the text, but I don't understand what was done from that sentence).

11) L277 – what phytoplankton bloomed in the system? Was it *Ostreococcus* ? Also, notably the phages that did not aerosolize are all to be expected marine phages, whereas those that did are not known in marine systems. Is this a contaminant signal ? also, P-GSP1 is a podovirus, but its clustering in fig 3 is oddly placed ...

12) L278 – there are now benchmark recommendations for assigning something to a particular virus from read-based analyses – see Roux et al 2017 PeerJ. Are you using these ? As well, if ocean viruses, why not map against the ocean virus genomes assembled from extensive ocean viral metagenomes now available. According to Roux et al 2016 Nature, surface ocean viruses are now sampled to saturation, at least at the genus level, so it is likely a good reference database for read-based mapping.

Selective transfer of bacteria and viruses from the ocean to the atmosphere

Response to Referees

Reviewer #1:

1. Lines 36 – 38. The data suggests an influence of surface properties, but how do the authors quantify the hydrophobic properties of the bacteria? Are they simply inferred from what is known of the cell envelope?

The hydrophobic properties are inferred from what is known of the cell envelope and supported by genomic information and patterns seen in the data. This data serves as a starting point to further examine more specific physiochemical properties that affect aerosolization. To remove ambiguity in the abstract we have edited the text at Line 36.

Original:

Line 36...The enrichment of mycolic acid-coated Corynebacteriales and lipid-enveloped viruses suggests hydrophobic properties increase transport to the sea surface and SSA.

Edit:

Line 36...Inferred from genomic comparisons, the enrichment of mycolic acid-coated Corynebacteriales and lipid-enveloped viruses suggests hydrophobic properties increase transport to the sea surface and SSA.

2. Lines 103 – 105. Why is there such a big difference in concentration? Wouldn't the amount of bacteria in the sample be independent of the distance to the collection ports? Do the authors expect that the difference is due to changes in cell viability? On a related point, how is the number of cells quantified?

In natural marine sampling, the aerosol concentration is quickly diluted by wind currents and measurements are generally many meters above the sea surface. We believe the higher concentrations of aerosols is due to the enclosed channel and the sampling ports proximity to the location to the wave break which is < 1 meter. The number of bacterial cells are determined by flow cytometry. We have added these details to the main text and moved the details of sampling port position from sample collection to the waveflume description in methods.

Original:

Line 101 Bulk chlorophyll *a* levels, cell counts, and viral counts were representative of natural oceanic bloom concentrations^{8,33,34} (Supplementary Information 1). Due to the proximity of the SSA collection ports to the breaking waves, the average aerosol bacterial concentrations ($1.26 \times 10^7 \pm 0.19 \times 10^7$ cells m^{-3}) were 1000-fold greater than those estimated in natural environments⁹.

Edit:

Line 107...Bulk chlorophyll *a* levels, cell counts by flow cytometry, and viral counts by epifluorescence microscopy were representative of natural oceanic bloom timescales and concentrations^{8,33,34} (Supplementary Note 3.1). Due to the proximity of the SSA collection ports to the breaking waves, the average aerosol bacterial concentrations ($1.26 \times 10^7 \pm 0.19 \times 10^7$ cells m^{-3}) were 1000-fold higher than those estimated in natural environments where aerosols are sampled many meters above the sea surface and diluted by the atmosphere⁹.

Line 360...Aerosol sampling ports were positioned ~3 ft downfield of the general position of the breaking waves.

3. Lines 151 – 153. It would seem that the shift in microbial population is a problem, how do we know that some of the species that disappeared wouldn't be preferentially released? Do the authors have hypotheses for why the population changes so quickly?

In many mesocosm and phytoplankton bloom studies, the population changes, biasing for species that thrive in the selected conditions and with the influx of nutrients. As shown in Teeling, H. *et al.* Substrate-controlled succession of marine bacterioplankton induced by a phytoplankton bloom. *Science*. **336**, 608–611 (2012), nutrient influxes that cause algal blooms also increase populations of Bacteroidetes, Alphaproteobacteria, Gammaproteobacteria similar to bacterial enrichments seen in our experiment and relevant to situations where large amounts of bacteria are generated and more relevant to airborne releases. However, biases owing to the adaptability of certain species to cultivation conditions is nearly impossible to avoid in synthetic growth systems. We believe that because aerosolization trends are shared across related taxa that many of these biases are minimized. We recognize that there are likely species abundant in marine ecosystems that were not present or did not thrive well in our system. Other studies will be needed to determine the aerosolization patterns of underrepresented species and additionally the different microbiomes of different marine ecosystems. However, the broad taxonomic patterns, such as the propensity for Actinobacteria to aerosolize, are likely to hold regardless of system. The main message to take from these studies that aerosolization in marine systems is not static and varies according to taxa. We have added more detail and clarification at Line 145.

Original:

Line 151...The aquatic microbial population shifted significantly from the original oceanic composition, but after Day 8 remained relatively constant for the duration of the experiment (Extended Data Fig. 1a).

Edit:

Line 145 ...Likely due to adaptation to the growth system and preference for species thriving in eutrophic conditions, the aquatic microbial population shifted significantly from the original oceanic composition, but after Day 8 remained relatively constant for the duration of the experiment with small shifts in individual taxa similar to bacterial community shifts observed during natural marine phytoplankton blooms^{39,42} (Supplementary Fig. 1a).

4. Lines 159 – 161. On a related point as above, how much does the microbiome in the ocean change typically in the amount of time relevant for these experiments?

In, Kim, H., Miller, A. J., McGowan, J. & Carter, M. L. Coastal phytoplankton blooms in the Southern California Bight. *Prog. Oceanogr.* **82**, 137–147 (2009). They show the month to month rapid vacillation of phytoplankton in a southwestern United States coastal zone with similar timescales to those seen in our experiment. Teeling, H. *et al.* (2012) also demonstrates the evolution of bacteria in response to an algal bloom with similar timescales and shifts in bacterial communities as seen in our experiment. We have added these important details to the text starting at lines 107 as also shown in comment 2 and 140 as also shown in the response to comments 2 and 6.

Original:

Line 101 Bulk chlorophyll *a* levels, cell counts, and viral counts were representative of natural oceanic bloom concentrations^{8,33,34} (Supplementary Information 1). Due to the proximity of the SSA collection ports to the breaking waves, the average aerosol bacterial concentrations ($1.26 \times 10^7 \pm 0.19 \times 10^7$ cells m^{-3}) were 1000-fold greater than those estimated in natural environments⁹.

Line 133 Phytoplankton bloom-associated copiotrophs—Alphaproteobacteria (Roseobacters), Flavobacteriia, and Gammaproteobacteria^{39–41}—were proportionately most abundant during the experiment, but comparisons between bulk, SSML, and aerosol community compositions reveal key differences (Fig. 1b).

Edit:

Line 107...Bulk chlorophyll *a* levels, cell counts by flow cytometry, and viral counts by epifluorescence microscopy were representative of natural oceanic bloom timescales and concentrations^{8,33,34} (Supplementary Note 3.1). Due to the proximity of the SSA collection ports to the breaking waves, the average aerosol bacterial concentrations ($1.26 \times 10^7 \pm 0.19 \times 10^7$ cells m^{-3}) were 1000-fold higher than those estimated in natural environments where aerosols are sampled many meters above the sea surface and diluted by the atmosphere⁹.

Line 140 While unbiased assessments of microbial diversity in isolated SSA to date are lacking, phytoplankton bloom-associated copiotrophs—Alphaproteobacteria (Roseobacters), Flavobacteriia, and Gammaproteobacteria^{39–41}—were abundant in bulk, SSML, and aerosol samples during the experiment. Comparisons between the taxa in these compartments reveal key differences in community distributions (Fig. 1b). Likely due to adaptation to the growth system and preference for species thriving in eutrophic conditions, the aquatic microbial population shifted significantly from the original oceanic composition, but after Day 8 remained relatively constant for the duration of the experiment with small shifts in individual taxa similar to bacterial community shifts observed during natural marine phytoplankton blooms^{39,42} (Supplemental Fig. 1a).

5. How does the simulated ocean work, hydrodynamically (e.g. wave generation and any mixing)? Would be helpful to see how the system mimics ocean hydrodynamic properties.

This work for this system is described elsewhere which is referenced in the text. The SSA is generated in a 33-m wave channel which reproduces the key physical production mechanisms of breaking waves in the ocean while allowing the minimization of background aerosol concentrations and using clean filtered air in the headspace to allow the detection of nascent SSA. Please see:

Prather, K. A., Bertram, T. H., Grassian, V. H., Deane, G. B., Stokes, M. D., Demott, P. J., Aluwihare, L. I., Palenik, B. P., Azam, F., Seinfeld, J. H., Moffet, R. C., Molina, M. J., Cappa, C. D., Geiger, F. M., Roberts, G. C., Russell, L. M., Ault, A. P., Baltrusaitis, J., Collins, D. B., Corrigan, C. E., Cuadra-Rodriguez, L. A., Ebben, C. J., Forestieri, S. D., Guasco, T. L., Hersey, S. P., Kim, M. J., Lambert, W. F., Modini, R. L., Mui, W., Pedler, B. E., Ruppel, M. J., Ryder, O. S., Schoepp, N. G., Sullivan, R. C. & Zhao, D. Bringing the ocean into the laboratory to probe the chemical complexity of sea spray aerosol. *Proc. Natl. Acad. Sci. U. S. A.* **110**, 7550–5 (2013).

This work outlines how this system generates sea spray aerosol representative of what is formed over oceans including relevant hydrodynamic data including details of wave generation. This novel system, to the best of our knowledge, is one of the only sources of isolated nascent SSA using filtered input air and with proper production mechanisms that capture the full bubble spectrum that occurs in natural breaking waves. As this publication detailing work to characterize this system was comprehensive, it was not possible to include these analyses in the manuscript. This work is referred to on Lines 75–84.

6. How do the taxa found in the aerosols compare with those from other studies?

The availability of metagenomic characterization of total bacteria including non-cultivable species from pure, isolated marine aerosols did not exist prior to these studies. Yooseph *et al.* 2009 gets the closest with metagenomic analysis of aerosols collected from a coastal location with obvious contamination from terrestrial sources. The taxa found in the aerosols in this experiment most resemble those found in seawater under copiotrophic conditions and followed the composition of the bulk and sea surface microlayer communities during the experiment. The lack of taxonomic characterization of aerosols is mentioned in the introduction, Line 60 and we have added more detail to make this clear in the text where the comparison to the composition of natural seawater is also offered.

Original:

Line 133 Phytoplankton bloom-associated copiotrophs—Alphaproteobacteria (Roseobacters), Flavobacteriia, and Gammaproteobacteria^{39–41}—were proportionately most abundant during the experiment, but comparisons between bulk, SSML, and aerosol community compositions reveal key differences (Fig. 1b).

Edit:

Line 140 While unbiased assessments of microbial diversity in isolated SSA to date are lacking, phytoplankton bloom-associated copiotrophs—Alphaproteobacteria (Roseobacters), Flavobacteriia, and Gammaproteobacteria^{39–41}—were abundant in bulk, SSML, and aerosol

samples during the experiment. Comparisons between the taxa in these compartments reveal key differences in community distributions (Fig. 1b). Likely due to adaptation to the growth system and preference for species thriving in eutrophic conditions, the aquatic microbial population shifted significantly from the original oceanic composition, but after Day 8 remained relatively constant for the duration of the experiment with small shifts in individual taxa similar to bacterial community shifts observed during natural marine phytoplankton blooms^{39,42} (Supplemental Fig. 1a).

7. Line 369 and Line 1: It seems that the title is too general, I think 'phytoplankton bloom' should be found in the title since the authors really explored a particular situation.

We have given the paper a more specific title as suggested:

Original:

Selective transfer of bacteria and viruses from the ocean to the atmosphere

Edit:

Taxon-specific aerosolization of bacteria and viruses in an experimental ocean-atmosphere mesocosm

8. Lines 545 – 546. Wouldn't you expect more realistic microbiome performance if you integrate light and dark cycles? Further, I expect that the aeration also influences the aerosol process and the stability/composition of the microbiome.

This is a good point. The more the conditions follow what occurs in nature the better the observations of the microbiome will be. The wave channel is a system that is evolving and light/dark cycling was not available at the start of the experiment. However, as we discussed above the biases to certain species in a synthetic system are difficult to avoid. The goal of this study was to determine whether differences in aerosolization existed based on taxonomy. Not only will natural systems have a different composition, different marine regions with different ecologies will additionally have a different community composition. The broader taxonomies present in these systems and in this study can be used together to identify relevant airborne species. We have added a discussion of this topic to the manuscript at Line 330.

Original:

Line 368... This study reveals the taxon-specific dynamics of viral and bacterial transfer across the air-sea interface during phytoplankton blooms.

Edit:

Line 330... This study provides a needed baseline for experimental marine microbial aerosolization studies that may now incorporate new variables such as diurnal light cycling, zooplankton grazing, different seed populations, and alternative nutrient amendments to further evaluate aerosolization dynamics of specific species and environments. The results reveal the taxon-specific dynamics of viral and bacterial transfer across the air-sea interface during phytoplankton blooms.

Reviewer #2:

1. Why was it impossible to count replicate samples for the viruses? Replication is critical for insuring good counts as these are notoriously problematic and variable.

Due to the small samples allowed by the parameters of the experiment and allotted to other analyses, there was not enough sample to perform these analyses in triplicate. This detail has been added to the methods. We note that the decreased aerosolization of viruses was also supported by the read-based analyses, which are discussed on Lines 224-231.

Original:

Line 569 ... It was not possible to perform aerosol virus counts on replicate samples.

Edit:

Line 397 ... It was not possible to perform aerosol virus counts on replicate samples due to the small sample sizes determined by the parameters of the experiment and volumes required by other analyses.

2. The methods need more details. For DNA extraction it is not clear whether the authors captured viral fraction samples from 2L of seawater or 200ml of SSML onto 0.02um filters. This would be nearly impossible, so if not what were the volumes captured on those filters? As well, how much DNA resulted? This is critical as amplification is commonly needed for viral metagenomes, but many amplification methods render the data non-quantitative, which would be problematic for most of the virus data presented in this paper. The informatics write-ups need to include versions and parameter settings for programs and dates for database access so that analyses could be repeated. Without this information it is not clear how to interpret the findings.

Thank you for pointing out this oversight as 0.025 μm filtered samples were performed on 1L samples for bulk and the rest as indicated. It is a keen observation that this was difficult and took long filtration times, and while the DNA yield were predominantly low (though on certain days had rather large spikes in accordance with DNA in the other sample types), we were able to obtain sufficient quantities for analysis likely due to the bloom conditions and the DNA extraction method which we found increased DNA yields and quality over the use of MoBio kits. We did not perform any amplification of the DNA prior to sequencing. We have added these details to methods. In addition, we have added DNA concentration data to the supplementary information (Supplemental Table 1). We have also added appropriate details to the informatics analyses in the methods section.

Original:

Line 573...Immediately after collection, samples were size fractionated by serially filtering on 3- μm polycarbonate filters (Whatman Nuclepore), 0.2- μm polyethylsulphone filters (Pall Supor. 200), and then 0.025- μm mixed cellulose filters (MF-Millipore.).

Line 599...Kraken was run in paired mode with the full NCBI RefSeq database.

Line 617...Sixty-two individual shotgun sequenced metagenomes were assembled separately using metaSPAdes¹⁴, with subsequent co-assembly of the assemblies using CLC (QIAGEN) to collapse redundant contigs. Genome quality throughout was assessed using QUAST¹⁵. Original reads for each sample were mapped to the contigs (CLC) to build sample specific coverage information. All contigs greater than 5 kbp were clustered into genome bins using VizBin¹⁶ and completeness of each of these bins was assessed using CheckM¹⁷ (Supplementary Data Table 2). Subsequently, 14 bins were recovered from a pangenome in the data by utilizing hierarchical clustering and sample specific coverage information. Genome annotations were determined utilizing APIS and manual curation¹⁸.

Edit:

Line 402...Immediately after collection, samples were size fractionated by serial filtration on 3- μ m polycarbonate filters (Whatman Nuclepore), then 0.2- μ m polyethylsulphone filters (Pall Supor 200), and then 0.025- μ m mixed cellulose filters (MF-Millipore). Full sample volumes (2 L bulk, 200 mL SSML, and 10 mL aerosol) were run through each filter with the exception of the 0.025- μ m bulk samples for which only 1L was filtered.... DNA was not amplified prior to sequencing.

Line 422...Kraken was run in paired mode with the full NCBI RefSeq database (release 70, April 30, 2015) using default cutoffs.

Line 447...Sixty-two individual shotgun sequenced metagenomes were assembled separately using SPAdes (v3.8.0, --meta mode, default kmer sizes of k21, 33, 55)⁶⁷, with subsequent co-assembly of the assemblies using CLC (QIAGEN) to collapse redundant contigs. Genome quality throughout was assessed using QUAST(v3.0) with default settings⁶⁸. Original reads for each sample were mapped to the contigs (CLC) to build sample specific coverage information. All contigs greater than 5 kbp were clustered into genome bins using VizBin⁶⁹ and completeness of each of these bins was assessed using CheckM (v1.0.5)⁷⁰ (Supplementary Table 4). Subsequently, 14 bins were recovered from a pangenome in the data by utilizing hierarchical clustering and sample specific coverage information. Genome annotations were determined utilizing APIS (<https://github.com/jhbagger/APIS>)⁷¹ and manual curation using PhyloDB (<https://scripps.ucsd.edu/labs/aallen/data/1.076>)⁷¹.

Added to Supplementary tables and Supplementary note 3.2.1.

3.2.1. DNA yields. DNA recovery and estimated original concentration in native samples are given in Supplementary Table 1. DNA abundance was determined by QubitTM (ThermoFisher).

Supplementary Table 1. DNA extraction yields.

Sample Date	Sample Type	Fraction	Total DNA recovered (ng)	Sample DNA concentration (ng/ml)*	Sample Date	Sample Type	Fraction	Total DNA recovered (ng)	Sample DNA concentration (ng/ml)*
7/3/14	Bulk	3µm	1914.00	1.91	7/27/14	Bulk	3µm	1152.00	5.76E-01
7/3/14	Bulk	0.2µm	2850.00	1.43	7/27/14	Bulk	0.2µm	636.00	3.18E-01
7/3/14	Bulk	0.025µm	73.44	1.47E-01	7/27/14	Bulk	0.025µm	335.20	3.35E-01
7/3/14	SSML	3µm	307.20	1.54	7/27/14	SSML	3µm	765.60	3.83
7/3/14	SSML	0.2µm	628.80	3.14	7/27/14	SSML	0.2µm	1195.20	5.98
7/3/14	SSML	0.025µm	2.60	1.30E-02	7/27/14	SSML	0.025µm	251.33	1.26
7/5/14	Aerosol	3µm	0.75	5.22E-07	7/27/14	Aerosol	3µm	3.20	1.44E-06
7/5/14	Aerosol	0.2µm	0.80	5.53E-07	7/27/14	Aerosol	0.2µm	21.76	9.07E-06
7/5/14	Aerosol	0.025µm	8.01	3.66E-06	7/27/14	Aerosol	0.025µm	2.72	1.10E-06
7/11/14	Bulk	3µm	554.40	2.77E-01	7/31/14	Bulk	3µm	991.20	4.96E-01
7/11/14	Bulk	0.2µm	567.60	2.84E-01	7/31/14	Bulk	0.2µm	1248.00	8.91E-01
7/11/14	Bulk	0.025µm	4.61	4.61E-03	7/31/14	Bulk	0.025µm	344.40	3.44E-01
7/11/14	SSML	3µm	244.80	1.22	7/31/14	SSML	3µm	456.00	2.28
7/11/14	SSML	0.2µm	342.40	1.71	7/31/14	SSML	0.2µm	624.00	3.12
7/11/14	SSML	0.025µm	71.34	3.57E-01	7/31/14	SSML	0.025µm	305.87	1.53
7/11/14	Aerosol	3µm	6.09	2.78E-06	7/31/14	Aerosol	3µm	0.91	3.80E-07
7/11/14	Aerosol	0.2µm	220.73	7.93E-05	7/31/14	Aerosol	0.2µm	2.81	1.17E-06
7/11/14	Aerosol	0.025µm	3.38	1.45E-06	7/31/14	Aerosol	0.025µm	1.41	5.87E-07
7/22/14	Bulk	3µm	559.20	2.80E-01	8/6/14	Bulk	3µm	583.20	2.92E-01
7/22/14	Bulk	0.2µm	1344.00	6.72E-01	8/6/14	Bulk	0.2µm	696.00	3.48E-01
7/22/14	Bulk	0.025µm	375.20	3.75E-01	8/6/14	Bulk	0.025µm	584.40	5.84E-01
7/22/14	SSML	3µm	274.40	1.37	8/6/14	SSML	3µm	122.64	6.13E-01
7/22/14	SSML	0.2µm	342.40	1.71	8/6/14	SSML	0.2µm	378.00	1.89
7/22/14	SSML	0.025µm	248.00	1.24	8/6/14	SSML	0.025µm	115.44	5.77E-01
7/22/14	Aerosol	3µm	3.02	1.26E-06	8/6/14	Aerosol	3µm	1.26	5.23E-07
7/22/14	Aerosol	0.2µm	5.53	2.30E-06	8/6/14	Aerosol	0.2µm	3.06	1.28E-06
7/22/14	Aerosol	0.025µm	2.59	1.20E-06	8/6/14	Aerosol	0.025µm	1.06	1.15E-06

*Bulk and SSML measurements are per mL of seawater, aerosol is per mL of air

3. All figures need to be legible – the fonts were far too small.

For space considerations, we did not send full resolution images with our original submission. We will work with editors to improve font visualization and increase font size as necessary.

4. In Fig 1a there are no viral count data, in spite of suggestion at L115 that there is. This is worth showing in the main text since a major conclusion is drawn from these data (that viruses are less aerosolized).

We have moved the viral counts data from Extended Data Fig 1c to included them in the main text denoted as Table 1.

Table 1 Virus particle (vp) counts by microscopy in bulk, SSML, and aerosol samples

Day	Bulk (x 10 ³) (vp mL ⁻¹)	Day	Bulk (x 10 ³) (vp mL ⁻¹)	Day	Bulk (x 10 ³) (vp mL ⁻¹)	Day	SSML (x 10 ³) (vp mL ⁻¹)	Day	SSML (x 10 ³) (vp mL ⁻¹)	Day	Aerosol (x 10 ³) (vp mL ⁻¹)	Day	Aerosol (x 10 ³) (vp mL ⁻¹)
4	1.86 ± 0.40	13	1.63 ± 0.34	23	4.76 ± 0.65	7	1.32 ± 0.19	22	3.04 ± 0.43	0	1.58 ± 0.18	14	1.32 ± 0.15
5	1.00 ± 0.29	14	1.44 ± 0.26	24	3.63 ± 0.52	8	1.21 ± 0.22	23	3.66 ± 0.35	1	0.70 ± 0.08	18	1.39 ± 0.15
6	2.17 ± 0.52	15	1.72 ± 0.33	25	8.01 ± 0.80	9	1.21 ± 0.22	24	3.96 ± 0.53	2	0.77 ± 0.09	20	0.33 ± 0.04
7	1.68 ± 0.32	16	2.99 ± 0.36	26	5.50 ± 0.74	10	1.43 ± 0.28	29	5.56 ± 0.92	4	1.01 ± 0.11	22	0.55 ± 0.06
8	0.60 ± 0.10	17	2.36 ± 0.26	27	5.07 ± 0.59	16	2.08 ± 0.27	32	1.59 ± 0.54	6	2.33 ± 0.26		
9	1.93 ± 0.21	18	2.91 ± 0.42	28	5.26 ± 0.62	17	3.54 ± 0.42	33	2.35 ± 0.54	7	1.04 ± 0.12		
10	1.10 ± 0.38	19	4.56 ± 0.43			18	3.85 ± 0.51	34	3.48 ± 0.68	9	2.14 ± 0.24		
11	1.01 ± 0.23	21	4.36 ± 0.36			20	3.66 ± 0.57	35	4.06 ± 0.49	11	1.14 ± 0.13		
12	1.24 ± 0.27	22	7.97 ± 0.79			21	3.90 ± 0.42	36	2.95 ± 0.34	13	1.20 ± 0.13		

5. In Fig 1c – what fraction of the reads were mapped to a reference genome? (reader is led to believe 100% with the plots shown) this is important for interpreting all of the downstream data presented on viruses.

These read based annotations in Figure 1c are reported at the taxonomic delineations of class and family where Kraken annotations should be reliable. For any analysis reporting species identity, spatial coverage has been conducted mapping these reads to a reference genome, which includes the phylogenetic analyses shown in Figs 3 and in Fig 4. Because many reads come from strains whose full genomes are not represented in the database, and because there is considerable range in relative abundance and hence genome coverage, only 88% of species above abundance thresholds in the data set containing 197 bacteria and 46 viruses. This was 28% of the complete abundance trimmed set. This has been clarified in the main text, methods, figure legends, and supplementary information.

Original:

Line 126 Spatial scaffold coverage of detected genomes was inspected, and genomes below a spatial coverage threshold (0.1%) were removed (Supplementary Information 2.2).

Line 607 ...Spatial coverage for each genome was calculated by implementing the following pipeline... The coverage trimmed data set omitted species below 0.1% spatial coverage.

Line 635... To ensure all abundant species were included, this was amended with the top 50 abundant species resulting in 197 bacteria species.

Supporting Information Section 2.2 Scaffolds from species' assignments were examined against published genomes (Supplementary Information Table 2). Species below 0.1% spatial coverage were removed resulting in a coverage trimmed set of 76 bacterial and 30 viral species.

Edit:

Line 133 For data reporting on species identity, additional spatial coverage analysis of 214 detected genomes with available full reference genomes was conducted. Genomes below a spatial coverage threshold (0.1% of total genome length) were removed (Supplementary Information 3.2.3).

Line 431 ...Spatial coverage analysis was performed to determine the percent of the genome that is represented in a reference genome by length. Spatial coverage for inspected genomes was calculated by implementing the following pipeline... 85% nucleotide identity is the mapping criteria. The availability of reference genomes allowed analysis of 214 of 243 genomes included in the trimmed set of 197 bacterial and 46 viral genomes. This represented 28% of the total genomes in the original abundance trimmed set (756 genomes). The coverage trimmed data set omitted species below 0.1% spatial coverage.

Line 471...To ensure all abundant species were included, this was amended with the top 50 abundant species resulting in 197 bacteria species. This set was further utilized for coverage analysis.

Supplementary Note 3.2.3. Scaffolds from species' assignments were examined against published genomes representing 214 of 243 inspected species (Supplementary Table 3). This represented 28% of the species above abundance thresholds. Species below 0.1% coverage were removed resulting in a coverage trimmed set of 76 bacterial and 30 viral species. Coverage trimming is applied to analyses reporting species identity.

6. L128 and L168 – not clear how complete a draft genome needs to be to be reported here, and how the cut-offs were chosen

We reported on all genome bins regardless of completeness. This data is shown in Supplementary Note 3.2.4, Supplementary Table 4, and has been noted in the methods.

Original:

Line 625...Genome annotations were determined utilizing APIS and manual curation¹⁸.

Edit:

Line 456...Genome annotations were determined utilizing APIS (<https://github.com/jhbagger/APIS>)⁷¹ and manual curation using PhyloDB (<https://scripps.ucsd.edu/labs/aallen/data/1.076>)⁷¹. All recovered genomes were reported regardless of completeness.

7) What is spatial scaffold coverage?

Spatial coverage is the percent of the reference genome that is mapped to. By imposing a spatial coverage cutoff, we exclude Kraken annotations that might result from common elements like transposons. We have added this rationale to the methods.

Original:

Line 607...Spatial coverage for each genome was calculated by implementing the following pipeline:...

Line 613...The coverage trimmed data set omitted species below 0.1% spatial coverage.

Edit:

Line 431...Spatial coverage analysis was performed to determine the percent of the genome that is represented in a reference genome. Spatial coverage for each genome was calculated by implementing the following pipeline:...

Line 441...The coverage trimmed data set omitted species below 0.1% spatial coverage to avoid erroneous annotations due common elements such as transposons (Supplementary Table 3).

8. L160 – where are the data that *E. coli* and *Legionella* were in the original seawater? This is a really important control to understand whether there are not contamination issues in the system.

We agree. While *Legionella* and *E. coli* species are shown in the original sample in Extended data. Fig 1a, it is difficult to see the presence of *Legionella*. We have added Supplementary Fig. 2, that shows the abundance of all draft genomes throughout the experiment and added this reference at line 155. This figure was in an earlier version of the manuscript that got omitted during editing. We apologize for discarding this vital piece of data.

Original:

Line 159...*E. coli* and *Legionella* were present in initial samples suggesting they were native to the coastal community but became enriched in our system during the experiment.

Edit:

Line 155...*E. coli* and *Legionella* were present in initial samples (Supplementary Fig. 2), suggesting they were native to the coastal community but became enriched in our system during the experiment.

Supplementary Fig. 2 Abundance of draft genomes. 18 bacterial and 6 viral draft genomes determined by assembly based methods. Heatmap indicates fractional abundance of species in the sample. Day of sample is indicated on top columns. For each 9-square group size fraction is indicated from left to right: >3 μm, 0.02 – 3 μm, and 0.025 – 0.2 μm, and top to bottom: Aerosol, SSML, and bulk water compartments. Schematic in upper right corner indicates this sample arrangement. Blanks indicate samples below threshold limits.

9. L168 – how many draft genomes were recovered?

We have amended the text to include this information.

Original:

Line 168 ...Draft genomes were recovered for several bacteriophage as well as an NCLDV (nucleoplasmic large DNA virus).

Edit:

Line 165 ... Draft genomes were recovered for five bacteriophage as well as an NCLDV (nucleoplasmic large DNA virus).

10. The main text needs to include some information on the quality of the metagenomes, as well as specific comparison of read-based vs assembly-based analyses. As the authors state in the results, Kraken does well where the reference genome database is strong, but its “accuracy is much lower in environmental samples”. This will be much worse for viruses since the databases are so bad. What are the cut-offs used to “map” a read to a reference genome? I would think utilizing the draft genomes assembled in main text figures would be needed to show this is a meaningful analysis since Kraken is not commonly used in environmental work (I see this is in L130 in the text, but I don’t understand what was done from that sentence).

85% nucleotide identity is the mapping criteria for the spatial coverage mapping. Kraken itself is a kmer mapper, and we use its default cutoffs. This has been added to the methods section (also shown with comments 2 and 5). The assemblies themselves were run through Kraken using the same pipeline as in read-based analyses. This detail has also been added to the methods section.

Original:

Line 625...Genome annotations were determined utilizing APIS and manual curation¹⁸.

Edit:

Line 456...Genome annotations were determined utilizing APIS and manual curation using PhyloDB (<https://scripps.ucsd.edu/labs/aallen/data/1.076>)³⁸. All recovered genomes were reported regardless of completeness. To assess agreement between assembly-based and read-based annotations, assemblies were additionally annotated using the same methods used in read-based analyses (Kraken with the NCBI Refseq database) (Supplementary Table 6).

*11. L277 – what phytoplankton bloomed in the system? Was it *Ostreococcus* ?*

We did not use a sequencing approach that was targeted to eukaryotes. However, the bloom was likely diatom dominated based on the recovery of a diatom chloroplast genome from the assembly. Additionally, non-quantitative microscopic examinations suggested the source of the bloom was a small centric diatom. This information has been added at Line 167.

Original:

Line 168...Draft genomes were recovered for several bacteriophage as well as an NCLDV (nucleoplasmic large DNA virus) (Extended Data Fig. 6, Supplementary Information 2.2).

Edit:

Line 165...Draft genomes were recovered for five bacteriophage as well as an NCLDV (nucleoplasmic large DNA virus) (Supplementary Table 4, 5). While analyses were not targeted toward eukaryotic genomes, the recovery of a draft genome for a diatom chloroplast and parallel microscopic observations of a small centric diatom suggest diatoms as the source of the phytoplankton blooms.

Also, notably the phages that did not aerosolize are all to be expected marine phages, whereas those that did are not known in marine systems. Is this a contaminant signal?

It is possible that the viruses captured on the air filters did not originate from the aquatic layers, through this is unlikely due to the position of air intake ports near the breaking waves, the enclosed nature of the system, and the high abundance of marine aerosols captured. Further, the same viruses are both observed in the aquatic and aerosol samples. Due to the likely unique coastal ecology and high human traffic in this area from which the water was sampled for the experiment, the coastal water is the likely source of the viruses.

also, P-GSP1 is a podovirus, but its clustering in fig 3 is oddly placed ...

We agree this placement is odd, but it is the output of the PhytoT pipeline (<http://phylot.biobyte.de/>). Ultimately, the topology of the tree is not the data that was generated in this study and is instead a vehicle used to present the data.

12. L278 – there are now benchmark recommendations for assigning something to a particular virus from read-based analyses – see Roux et al 2017 PeerJ. Are you using these? As well, if ocean viruses, why not map against the ocean virus genomes assembled from extensive ocean viral metagenomes now available. According to Roux et al 2016 Nature, surface ocean viruses are now sampled to saturation, at least at the genus level, so it is likely a good reference database for read-based mapping.

While we recognize the value of these recommendations, they were published after we concluded the analyses, and we were not able to utilize them in our study. A more thorough examination of using newly available curated databases and accepted policies is an ideal follow-up and warrants a stand-alone study. We also acknowledge that the work referenced here in represent a big step forward in viral genomics and reference Roux et al 2016 at Line 160.

Original:

Line 165 *Podoviridae* and *Myoviridae* were the most common viruses identified in raw read assignments, and the viral population varied across the air–sea interface (Fig. 1c).

Edit:

Line 160 *Podoviridae* and *Myoviridae* were the most common viruses identified in raw read assignments, and the viral population varied across the air–sea interface (Fig. 1c). Despite recent advances in characterizing marine viral genomes⁴⁵ (Roux et al 2016), fewer viral genomes were recovered compared to bacterial genomes due to under-developed viral databases.

REVIEWERS' COMMENTS:

Reviewer #1 (Remarks to the Author):

I have reviewed the authors' response to the comments and changes to the manuscript and believe they now address all of my concerns sufficiently.

Reviewer #3 (Remarks to the Author):

Overall, I found that the authors adequately addressed most of reviewer #2 questions and issues. The only part I think is still lacking in detail is the methods section describing the bioinformatics analysis. In my opinion, the current level of detail regarding especially the annotation and taxonomic affiliation of genomes is not comprehensive enough for another researcher to faithfully reproduce the analysis performed. This is relatively easy to fix though, by simply explaining in a few sentences how does the APIS tool work (especially since the URL provided <https://github.com/jhbagger/APIS> does not seem to be active anymore) and which databases, search methods, and cutoffs were applied by the authors.

Specific comments:

Reviewer #2. Q1. I understand that replication could not be always achieved for virus counts due to sampling limitations, and this is now clarified in the text. However, I would suggest the authors to revise the text l. 227-228 when interpreting the decrease in relative abundance of viral reads: this could come from a decreased viral transfer or a lower infection rate in the aerosol samples. Ultimately, both parameters are linked, but I find the language used by the authors "demonstrating decreased transfer" a little too strong and would prefer e.g. "suggesting decreased transfer".

Q2. I thank the authors for expanding the methods section. For a reader to be able to reproduce the analysis, I found that the annotation section is still lacking in details, especially since the link provide for APIS (<https://github.com/jhbagger/APIS>) seems to be non-functioning. The methods section should at least briefly explain how this annotation software functions, i.e. which program was used to identify sequence similarity, which database was used as reference (and which version / date). For example, does APIS use the same PhyloDB database as was used in the manual curation step ?

Q3. Ok.

Q4. I agree with the authors that getting the virus counts in Table 1 help highlight this important information. I would still suggest to (i) only cite Table 1 l. 122 instead of "Fig.1A, Table1", since it looks otherwise as there should be some virus data in Fig. 1A, and (ii) refer to Table 2 l. 124 (after "preferential aerosolization of bacteria relative to viruses"), to help readers wrap their minds around these data.

Q5-6-7-8-9-10-11-12. I thank the authors for the clarifications and edits made to the text.

l. 120 and in figures:

I believe Vp should be VLP, i.e. virus-like particles

l. 233: "distributions were shifted more negative" I believe this should be "more negatively" ?

l. 253: "Similar to bacteria, select genomes differed from their closest relatives in their aerosolization behavior.". I'm confused by that, as I believe the opposite was stated for the bacterial fraction l. 203

“Closely related species have nearly identical aerosolization patterns that are propagated across [...]”.
Please clarify

I. 257: I believe “bacteriophage” should be “bacteriophages”

RESPONSE TO REVIEWERS' COMMENTS:

The authors would like to thank the reviewers for their helpful suggestions and comments. Responses to individual comments are below original reviewer comments (italicized). Additionally, edits made accordingly are highlighted in the text using track changes.

Reviewer #1 (Remarks to the Author):

I have reviewed the authors' response to the comments and changes to the manuscript and believe they now address all of my concerns sufficiently.

Reviewer #3 (Remarks to the Author):

Overall, I found that the authors adequately addressed most of reviewer #2 questions and issues. The only part I think is still lacking in detail is the methods section describing the bioinformatics analysis. In my opinion, the current level of detail regarding especially the annotation and taxonomic affiliation of genomes is not comprehensive enough for another researcher to faithfully reproduce the analysis performed. This is relatively easy to fix though, by simply explaining in a few sentences how does the APIS tool work (especially since the URL provided <https://github.com/jhbagger/APIS> does not seem to be active anymore) and which databases, search methods, and cutoffs were applied by the authors.

This is addressed in the response to Reviewer #2, Question 2, below.

Specific comments:

Reviewer #2. Q1. I understand that replication could not be always achieved for virus counts due to sampling limitations, and this is now clarified in the text. However, I would suggest the authors to revise the text l. 227-228 when interpreting the decrease in relative abundance of viral reads: this could come from a decreased viral transfer or a lower infection rate in the aerosol samples. Ultimately, both parameters are linked, but I find the language used by the authors “demonstrating decreased transfer” a little too strong and would prefer e.g. “suggesting decreased transfer”.

We replaced “demonstrating” with “suggesting” as requested at line 238.

Q2. I thank the authors for expanding the methods section. For a reader to be able to reproduce the analysis, I found that the annotation section is still lacking in details, especially since the link provide for APIS (<https://github.com/jhbagger/APIS>) seems to be non-functioning. The methods section should at least briefly explain how this annotation software functions, i.e. which program was used to identify sequence similarity, which database was used as reference (and which

version / date). For example, does APIS use the same PhyloDB database as was used in the manual curation step ?

We are grateful that you tested the link which unfortunately contained an error rendering the link invalid. The correct link, <https://github.com/jhbadger/APIS>, has been corrected in the text at line 469. We apologize for the mistake. Additionally, we have amended this section for added clarity additionally shown below (line 462) and provided an additional reference, 72 (<https://www.ncbi.nlm.nih.gov/pmc/articles/PMC3379637/>).

“Genome annotations were determined utilizing APIS (<https://github.com/jhbadger/APIS>)⁷¹ and manual curation using PhyloDB (<https://scripps.ucsd.edu/labs/aallen/data/>)⁷¹ as the reference database. APIS automates the process of sequence similarity, alignment, and phylogenetic inference for each protein in a given data set.⁷² APIS results were manually curated to determine predicted bin level taxonomy.”

Q3. Ok.

Q4. I agree with the authors that getting the virus counts in Table 1 help highlight this important information. I would still suggest to (i) only cite Table 1 l. 122 instead of “Fig. 1A, Table 1”, since it looks otherwise as there should be some virus data in Fig. 1A, and (ii) refer to Table 2 l. 124 (after “preferential aerosolization of bacteria relative to viruses”), to help readers wrap their minds around these data.

To clarify this section, we moved the reference to Fig. 1a to be with “cell” to make it clear that the reference is for bacteria only. This is at line 129, “In seawater, virus concentrations are generally 5 to 25 times greater than bacteria³⁵, and here we observed 10-fold greater bulk viral concentration by cell (Fig. 1a) and particle counts (Table 1).”

We also added in a reference for Table 2 at the location indicated at line 132.

Q5-6-7-8-9-10-11-12. I thank the authors for the clarifications and edits made to the text.

l. 120 and in figures:

I believe Vp should be VLP, i.e. virus-like particles

This has been changed in the throughout manuscript as shown in Table 1, and in the text at line 127.

l. 233: “distributions were shifted more negative” I believe this should be “more negatively” ?

“negative” edited to “negatively” in the text at line 238.

l. 253: “Similar to bacteria, select genomes differed from their closest relatives in their aerosolization behavior.”. I’m confused by that, as I believe the opposite was stated for the bacterial fraction l. 203 “Closely related species have nearly identical aerosolization patterns that are propagated across [...]”. Please clarify

Yes, this distinction is important. We have clarified the text at line 262. The sentence has been changed to “Similar to bacteria, while most viral genomes followed the aerosolization patterns of their taxa, select genomes differed from their closest relatives in their aerosolization behavior revealing genomic differences that may be useful in determination of relevant biochemical pathways.”

l. 257: I believe “bacteriophage” should be “bacteriophages”

“bacteriophage” has been changed to “bacteriophages at line 270.